# N-Terminal Pro-Brain Natriuretic Peptide Levels Are Associated with Post-Stroke In-Hospital Complications

**DOI:** 10.3390/jpm13030474

**Published:** 2023-03-05

**Authors:** María Luisa Ruiz-Franco, Eva Guevara-Sánchez, Laura Amaya-Pascasio, Miguel Quesada-López, Antonio Arjona-Padillo, Juan Manuel García-Torrecillas, Patricia Martínez-Sánchez

**Affiliations:** 1Stroke Unit, Department of Neurology, Torrecárdenas University Hospital, 04009 Almería, Spain; 2Department of Emergency Medicine, Torrecárdenas University Hospital, 04009 Almería, Spain; 3Biomedical Research Unit, Hospital Universitario Torrecárdenas, 04009 Almería, Spain; 4Instituto de Investigación Biomédica Ibs. Granada, 18012 Granada, Spain; 5Centro de Investigación Biomédica en Red de Epidemiología y Salud Pública (CIBERESP), 28029 Madrid, Spain

**Keywords:** atrial fibrillation, stroke, young, severity, outcomes

## Abstract

Previous studies have shown the relationship between N-terminal pro-brain natriuretic peptide (NT-proBNP) with stroke mortality and functional outcome after an acute ischemic stroke (AIS). Knowledge of its association with systemic and neurological in-hospital complications is scarce. Our objective is to analyze this. We performed an observational, retrospective study that included consecutive AIS patients during a 1-year period (2020). A multivariate analysis was performed to identify if NT-proBNP levels were independently associated with in-hospital complications. 308 patients were included, of whom 96 (31.1%) developed systemic and 62 (20.12%) neurological in-hospital complications. Patients with any complication (39.3%) showed higher NT-proBNP levels than those without (median (IQR): 864 (2556) vs. 142 (623) pg/dL, *p* < 0.001). The receiver operating characteristic curve (ROC) pointed to 326 pg/dL of NT-proBNP as the optimal cutoff level for developing in-hospital systemic complications (63.6% sensitivity and 64.7% specificity for any complication; 66.7% and 62.7% for systemic; and 62.9% and 57.7% for neurological complications). Multivariate analyses showed that NT-proBNP > 326 pg/dL was associated with systemic complications (OR 2.336, 95% CI: 1.259–4.335), adjusted for confounders. This did not reach statistical significance for neurological complications. NT-proBNP could be a predictor of in-hospital systemic complications in AIS patients. Further studies are needed.

## 1. Introduction

In-hospital complications are major causes of death in acute ischemic stroke (AIS) patients [1,2,3,4]. Currently, the prognosis of patients with AIS remains a challenge due to the limited existence of predictive studies. In this scenario, blood-based biomarkers might provide additional information to established prognostic factors [5].

Brain natriuretic peptide is a cardiac hormone produced from the cleavage of its precursor pro-BNP and is secreted together with a biologically inactive terminal fragment, the N-terminal pro-brain natriuretic peptide (NT-proBNP) [6]. These two molecules are important predictive and prognosis biomarkers for cardiovascular disease, especially NT-proBNP, due to its longer half-life, higher plasma concentration and better diagnostic resolution [7].

Previous studies have shown that plasma of NT-proBNP are elevated in two thirds of patients with AIS [8]. This may be because atrial fibrillation (AF), as well as cardiac dysfunction in general, is associated with increased levels of NT-proBNP [9,10,11]. Moreover, NT-proBNP levels have been related with the appearance of acute decompensated heart failure (DHF) and respiratory tract infections (RTI) in AIS patients [5].

There is a lack of studies examining the relationship between blood levels of NT-proBNP and in-hospital complications after stroke, both systemic and neurological. Our hypothesis is that higher NT-proBNP levels could be predictor of a higher risk of in-hospital complications after an AIS. Our objective was to analyze this.

## 2. Materials and Methods

A retrospective analysis of consecutive AIS patients admitted to the Torrecárdenas University Hospital (TUH, Almería, Spain) during 2020 was conducted. TUH is the referral Stroke Center for a population of 739,293 inhabitants. Patients in whom NT-proBNP determination could not be performed were excluded from the study.

All patients were initially treated in the emergency room by a neurologist. Each patient was examined according to a standard protocol that included routine laboratory analyses, chest X-rays, ECG and neuroimaging.

The variables included were: (a) demographic data (sex and age); (b) pre-stroke functional status by modified Rankin Scale (mRS) score, classifying patients in two groups: mRS lower than 2 and greater than or equal to 2; (c) vascular risk factors: arterial hypertension (defined as having a previous clinical diagnosis of arterial hypertension or regular treatment with antihypertensive drugs or two or more prestrike readings above 140 and/or 90 mm Hg), dyslipidemia (previous diagnosis and/or current treatment with lipid-lowering drugs), diabetes mellitus (previous diagnosis and/or current treatment with insulin or oral hypoglycemic medications), smoking, alcohol abuse, previous atrial fibrillation, coronary heart disease (angina pectoris or myocardial infarction), valvular heart disease (previous diagnosis with echocardiography or diagnosis during the hospitality study), renal dysfunction (previous diagnosis of renal failure or diabetic or hypertensive nephropathy) and previous stroke; (d) prior treatments: antiplatelet agents, anticoagulants, statins, antihypertensives, oral antidiabetics and insulin; (e) stroke severity evaluated by the National Institutes of Health Stroke Scale (NIHSS); (f) acute phase treatments: thrombolysis and mechanical thrombectomy; (g) stroke etiology classified by TOAST classification; (h) stroke unit length of stay; (i) blood pressure on admission (diastolic, systolic and pulse pressure); (j) biochemical markers in peripheric blood tests: NT-proBNP (pg/mL), creatinine (mg/dL), glycaemia (mg/dL); k) systemic complications and (l) neurological complications:− Systemic complications: RTI (2 symptoms: dyspnea, fever, or purulent expectoration and one sign: specific signs in lung auscultation or in a chest X-ray), other infections (urinary tract infection −1 symptom: dysuria or high urinary frequency and one sign: choluria, alterations in the urine analysis such as nitrites or urine culture, sepsis, change in mental status, systolic blood pressure less than or equal to 100 mm of mercury, respiratory rate higher than or equal to 22 breaths per minute, cutaneous herpes, compatible cutaneous lesion with response to adequate treatment, phlebitis, inflammation signs with or without fever and with adequate response to the venous line exchange, febrile syndrome, fever and symptoms such as headache, chills or muscle and joint pains) and other non-infective complications (DHF—defined as a new onset or rapidly or gradual worsening of heart failure symptoms that require urgent therapy, acute urinary retention and gout attack).− Neurological complications: neurological impairment (worsening of previous cognitive level), hemorrhage (diagnosed with cerebral scan or magnetic resonance), malignant edema (diagnosed with cerebral scan or magnetic resonance with midline shift) or seizures (seizures after the stroke in patients without a previous history of epilepsy).

Diagnosis of in-hospital complications was made by the physician in charge of the patient, who was blinded to the conduct of this study. The physician in charge was always a stroke neurologist, who clinically assessed the patient daily and checked for in-hospital complications, in accordance with the stroke unit’s protocol. In addition, serum NT-proBNP determination was performed in the central laboratory of the hospital by technicians who were also blinded to the objectives of this study.

### 2.1. NT-ProBNP Measurements

A peripheral blood sample was taken within 48 h of hospitalization and serum NT-proBNP was measured by a central laboratory on the Roche Elecsys assay (analytical range 10–35,000 pg/mL).

### 2.2. Statistical Analysis

Patients were divided into two groups according to the presence of in-hospital complications, as well as whether these were systemic or neurological.

For the quantitative variables, the mean and standard deviation or the median and the interquartile range, were calculated and, for qualitative data, the absolute and relative frequency was offered. Comparisons between the groups were carried out using parametric or non-parametric tests. In case of homoscedasticity and normality of the samples, *t*-test was used for comparing quantitative variables, and the Mann-Whitney test in the case of non-normal data, for dichotomous variables. ANOVA and Kruskal-Wallis tests were used for comparing quantitative variables, in case of polytomous variables, when appropriate. Furthermore, post-hoc analyses, such as Bonferroni test or Kruskal-Wallis H test were carried out to test differences between groups.

Comparisons between groups were analyzed with the χ^2^ or Fisher’s exact test for dichotomous variables. Continuous variables were expressed as mean ± SD or median (interquartile range) and compared with Student’s *t* test or the Mann-Whitney test, as appropriate. The relationship between NT-proBNP levels and the presence of in-hospital complications was analyzed in three steps. First, a receiver operating characteristic (ROC) curve analysis was conducted to determine the predictive value of the area under the curve (AUC), as well as a cutoff point for NT-proBNP levels. We considered the point at which the sum of the specificity and sensitivity was highest, giving the same weight to false-positives and false-negatives. Second, the relationship between this NT-proBNP cutoff point and the developments of in-hospital complications was assessed using multivariate logistic regression models. Variables with a value of *p* ≤ 0.2 in the univariate testing were included, as well as those variables which are known to be associated with in-hospital complications and NT-proBNP levels (age, blood pressure, renal dysfunction, atrial fibrillation and ischemic heart disease). A backward procedure was followed as the modeling strategy, using the log-likelihood ratio test to assess the correctness of fit and compare nested models. Those variables that, when eliminated, produced a change of ≥15% in the OR (Odds Ratio) were considered as confounding variables. We conducted another ROC curve with the multivariate analysis results. Third, a forward stepwise (likelihood ratio) multivariate logistic analysis that included the variables selected by the backward analysis was developed to test the stability of the model. 

Ninety-five percent confidence intervals (CI) are presented. All tests were two-sided and *p* values of 0.05 or less were considered statistically significant. Statistical analysis was performed using IBM SPSS Statistics for Windows, Version 25.0, software (IBM Corp. Armonk, NY, USA) for Windows.

The study was approved by the local ethics committee and patient consent was not sought due to the retrospective nature of the study. The data that support the findings are available from the corresponding author on reasonable request.

## 3. Results

During the study period, a total of 476 stroke patients were admitted to the Stroke Center. Some were excluded from the study for the following reasons: cerebral hemorrhage (70), transient ischemic attack (18), venous infarction (7), patients did not meet the criteria for admission to the Stroke Unit (28), and incomplete data (45). A total of 308 patients were finally included in the analysis (Figure 1). Of these, 96 (31.1%) developed systemic complications, 62 (20.12%) neurological and 121 (39.3%) developed any complication (systemic or neurological).

Table 1 shows the baseline and clinical characteristics of AIS patients according to the presence of in-hospital complications. Patients with any complication (39.3%) showed higher NT-proBNP levels than those without (median (IQR): 864 (2556) vs. 142 (623) pg/dL, *p* < 0.001). Figure 2 shows the distribution of NT-proBNP levels according to different complications. NT-proBNP levels were significantly increased in RTI, DHF and hemorrhagic transformation.

The receiver operating characteristic curve (ROC) pointed to 326 pg/dL of NT-proBNP as the optimal cutoff level for developing systemic complications during hospitalization (63.6% sensitivity and 64.7% specificity for any complication; 66.7% and 62.7% for systemic; and 62.9% and 57.7% for neurological complications, respectively) (Figure 3).

The multivariate analyses (Table 2) showed that NT-proBNP > 326 pg/dL was associated with systemic complications (OR 2.336, 95% CI: 1.259–4.335), adjusted for confounders. However, it did not reach statistical significance for neurological complications.

## 4. Discussion

To our knowledge, this is the first study to analyze the association between blood NT-proBNP levels and post-stroke in-hospital complications, both systemic and neurological, in a cohort of unselected patients with cerebral infarction. Our results suggest that this biomarker is associated with a doubling of the likelihood of systemic complications, although its relationship with neurological complications is unclear.

The use of blood biomarkers for the prediction of complications after ischemic stroke has been widely studied in recent years [5]. NT-proBNP levels are increased in the acute phase of AIS in patients with and without cardiac failure [12]. Therefore, NT-proBNP has been used in many models to predict mortality after stroke alone [13,14,15] or in combination with other biomarkers such as NIHSS scores [16] or troponin T levels [17,18]. A recent meta-analysis of five studies found that NT-proBNP levels were associated with functional outcomes [10].

The results of our study support the use of NT-proBNP levels as a biomarker to predict the occurrence of in-hospital complications in AIS. This could be explained by the ability of the damaged brain to influence the heart through a local and systemic inflammatory response and a dysregulated autonomic activity influenced by the hypothalamic–pituitary–adrenal axis [19]. Some previous studies show that lesions affecting the insular cortex are in relation with a higher risk of blood pressure variations, arrhythmias, and myo-cytolysis [20,21,22]. Moreover, left hemisphere brain infarction associates risk of adverse cardiac outcomes and increased long-term mortality [23]. AIS also induces a catecholaminergic storm with the consequence of cardiomyocyte necrosis, fibrosis, and cardiac arrhythmias (such as AF), which would probably lead to cardiac dysfunction and increased NT-proBNP [24].

Other short-term and long-term complications that could be predicted with NT-ptoBNP levels have been studied previously. One study showed that the risk of recurrent stroke was in relation to admission BNP levels [25]. Another study demonstrated that NT-proBNP levels were significantly higher in patients with larger infarcts, higher mRS scores, and higher CHADS2 scores [26]. Of course, as it is a cardiac biomarker, it has been widely stablished in several studies that higher levels of NT-proBNP are predictors of cardioembolic etiology and previous or new FA [27,28,29]. Interestingly, recent studies have shown that NT-proBNP is related to aging, suggesting that it could be used to estimate true biological age or “proBNP age” and could be an end point surrogate in epidemiological and clinical studies [30]. This would explain why, in the multivariate models of in-hospital complications carried out in the present study, age is not a factor independently associated with in-hospital complications, while NT-proBNP is.

The relationship between RANKL gen polymorphins has been recently demonstrated as an independent risk factor for ischemic stroke in a study based on Italian population [31]. It may be interesting to analyze its influence in NT-proBNP levels in further studies.

In this study, the bivariate analysis showed that NT-proBNP levels were associated with a higher frequency of neurological complications, especially hemorrhagic transformation. This association is diluted in the multivariate analysis, which could be due to the lower weight of this biomarker in the development of neurological complications and the small sample size. Indeed, a recent study has shown that NT-proBNP levels are independently associated with hemorrhagic transformation, poor functional outcomes, and 3-month mortality in stroke patients who underwent intravenous thrombolysis [32]. Furthermore, in another large (N = 1039) observational study with stroke patients who received reperfusion therapies, elevation of Ln(NT-proBNP) was independently associated with malignant edema [28].

Our study has several limitations. Firstly, it is a single-center retrospective observational study, although data have been included prospectively. Secondly, the sample size remains relatively small, although it is one of the largest case series published. Finally, the sample included subjects with previous AF and ischemic heart disease, which could produce variations in NT-proBNP values due to their cardiac and non-neurological pathology.

Future studies are warranted to establish the role of NT-proBNP levels in the development of post-stroke complications. If a relevant role is confirmed, the inclusion of its measurement in stroke management guidelines could be recommended, since predicting the appearance of complications would help optimize resources and anticipate future clinical complications.

## 5. Conclusions

In conclusion, NT-proBNP could be a predictor of in-hospital systemic complications in ischemic stroke patients. Its determination in all stroke patients would be useful in order to prevent and treat complications early. Further studies are needed to clarify the role of NT-proBNP as a prognostic marker in these patients.

## Figures and Tables

**Figure 1 jpm-13-00474-f001:**
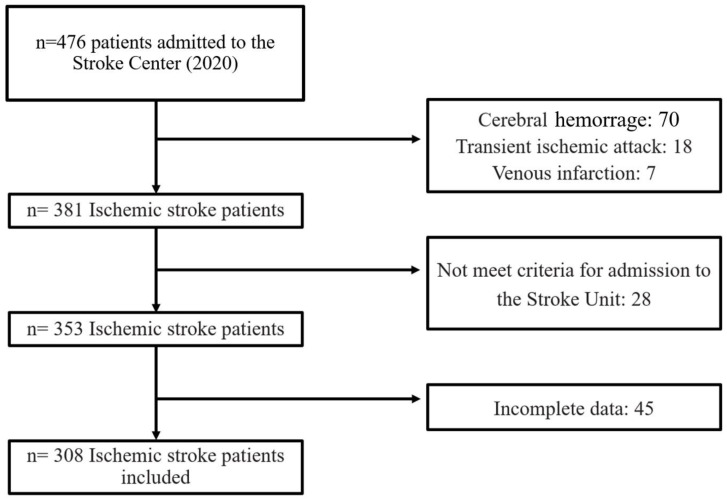
Study flowchart.

**Figure 2 jpm-13-00474-f002:**
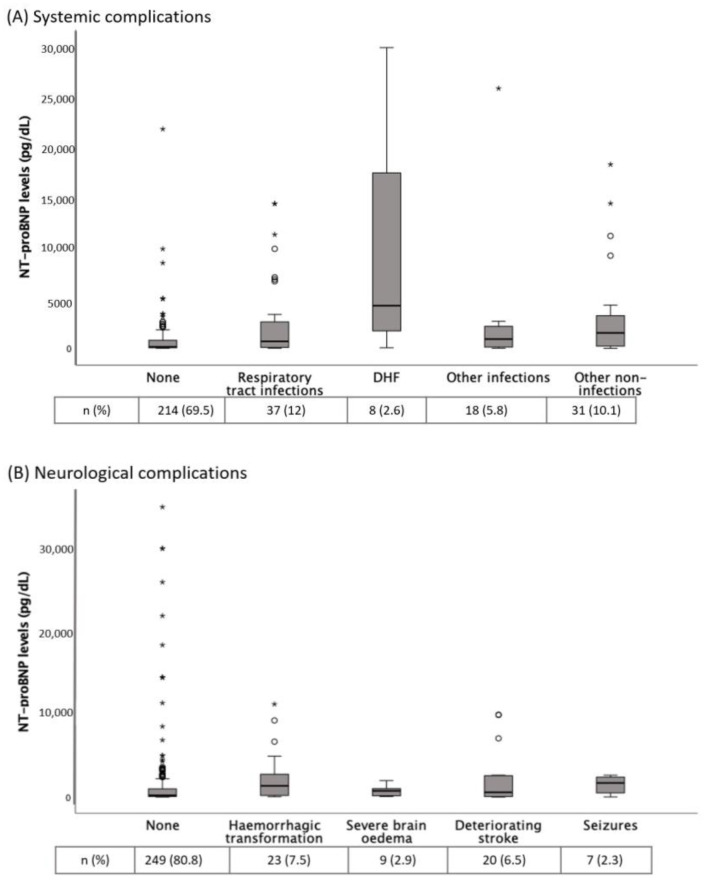
(**A**) NT-proBNP levels according to different systemic complications. *p* = 0.012 for comparison between no complication and respiratory tract infections; *p* = 0.01 for comparison between no complication and other non-infections groups; *p* = 0.002 for comparison between no complication and decompensated heart failure (DHF); *p* = NS for the other comparisons. (**B**) NT-proBNP levels according to different neurological complications. *p* = 0.029 for comparisons between no complication and hemorrhagic transformation; *p* = NS for the other comparisons. Tables show the complications frequency. The Kruskal-Wallis H test was used for between-groups comparisons. * NT-proBNP, N-terminal fragment of the prohormone of brain natriuretic peptide.

**Figure 3 jpm-13-00474-f003:**
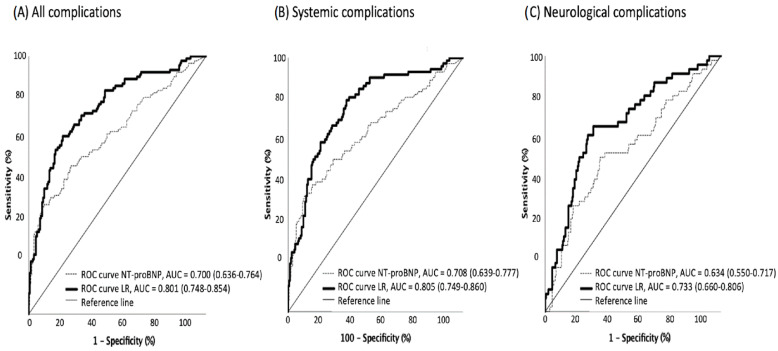
ROC curves for in-hospital complications based on NT-proBNP levels, alone and adjusted. ROC curves for all in-hospital complications (**A**), systemic complications (**B**) and neurological complications (**C**) in patients with ischemic stroke, based on NT-proBNP levels (dotted line) and on the logistic regression (LR) models (continuous line) described in Table 2.

**Table 1 jpm-13-00474-t001:** Baseline and clinical characteristics of acute ischemic stroke patients according to the presence of in-hospital complications.

	All Complications			Systemic Complications			Neurological Complications		
	Yes (n = 121)	No (n = 187)	*p* Value	Yes (n = 96)	No (n = 212)	*p* Value	Yes (n = 62)	No (n = 246)	*p* Value
Demographic data and comorbidities									
Male sex, n (%)	77 (63.6)	137 (73.3)	0.073	60 (62.5)	154 (72.6)	0.073	38 (61.3)	176 (71.5)	0.117
Age, median (IQR), years	74 (26)	59 (19)	<0.001	74 (26)	60 (21)	<0.001	73 (26)	63 (22)	<0.001
Previous mRS < 2, n (%)	93 (76.9)	168 (89.9)	0.002	73 (76%)	188 (88.7)	0.004	48 (77.4)	213 (86.6)	0.073
Arterial hypertension, n (%)	68 (56.2)	108 (57.8)	0.788	52 (54.2)	124 (58.5)	0.478	34 (54.8)	142 (57.7)	0.682
Dyslipidemia, n (%)	54 (44.6)	71 (38)	0.245	45 (46.9)	80 (37.7)	0.130	28 (45.2)	97 (39.4)	0.412
Diabetes mellitus, n (%)	37 (30.6)	40 (21.4)	0.069	39 (31.3)	47 (22.2)	0.088	15 (24.2)	62 (25.2)	0.870
Smoking, n (%)	47 (38.8)	98 (52.4)	0.020	38 (39.6)	107 (50.5)	0.076	22 (35.5)	123 (50)	0.041
Alcohol abuse, n (%)	14 (11.6)	36 (19.3)	0.074	11 (11.5)	39 (18.4)	0.126	5 (8.1)	45 (18.3)	0.051
Atrial fibrillation, n (%)	31 (25.6)	22 (11.8)	0.002	26 (27.1)	27 (12.7)	0.002	17 (27.4)	36 (14.6)	0.017
Coronary heart disease, n (%)	21 (17.4)	15 (8)	0.013	17 (17.7)	19 (9)	0.027	8 (12.9)	28 (11.4)	0.739
Valvular Heart Disease, n (%)	0 (0)	6 (3.2)	0.085	0 (0)	6 (2.8)	0.182	0 (0)	6 (2.4)	0.604
Renal dysfunction, n (%)	27 (23.1)	19 (10.6)	0.004	25 (26.6)	21 (10.3)	<0.001	9 (15.5)	37 (15.5)	0.995
Previous stroke, n (%)	21 (17.4)	21 (11.3)	0.131	14 (14.6)	28 (13.3)	0.756	11 (17.7)	31 (12.7)	0.298
Prior treatments									
Antiplatelet Agents, n (%)	15 (12.4)	21 (11.2)	0.756	13 (13.5)	23 (10.8)	0.496	5 (8.1)	31 (12.6)	0.320
Anticoagulants, n (%)	15 (12.4)	11 (5.9)	0.045	13 (13.5)	13 (6.1)	0.030	8 (12.9)	18 (7.3)	0.157
Statins, n (%)	41 (33.9)	54 (28.9)	0.353	31 (32.3)	64 (30.2)	0.711	24 (38.7)	71 (28.9)	0.133
Antihypertensives, n (%)	70 (57.9)	89 (47.6)	0.079	54 (56.3)	105 (49.5)	0.274	37 (59.7)	122 (49.6)	0.156
Oral antidiabetics, n (%)	15 (12.4)	14 (7.5)	0.154	14 (14.6)	15 (7.1)	0.037	4 (6.5)	25 (10.2)	0.371
Insulin, n (%)	9 (7.4)	9 (4.8)	0.337	9 (9.4)	9 (4.2)	0.075	2 (3.2)	16 (6.5)	0.544
Stroke severity									
NIHSS, median (IQR)	9 (12)	3 (5)	<0.001	10 (12)	3 (5)	<0.001	8 (12)	4 (7)	<0.001
Acute phase treatments									
Thrombolysis, n (%)	35 (28.9)	41 (21.9)	0.164	27(28.1)	49 (23.1)	0.345	20 (32.3)	56 (22.8)	0.121
Mechanical thrombectomy, n (%)	30 (24.8)	24 (12.8)	0.007	26 (27.1)	28 (13.2)	0.003	11 (17.7)	43 (17.5)	0.961
Stroke etiology (TOAST), n (%)									
Large-vessel occlusive	25 (20.7)	31 (16.6)	<0.001	20 (20.8)	36 (17)	<0.001	12 (19.4)	44 (17.9)	0.014
Small-vessel occlusive	10 (8.3)	58 (31)		8 (8.3)	69 (28.3)		4 (6.5)	64 (26)	
Cardioembolic	50 (41.3)	47 (25.1)		42 (43.8)	55 (25.9)		27 (43.5)	70 (28.5)	
Other	10 (8.3)	15 (8)		8 (8.3)	17(8)		5 (8.1)	20 (8.1)	
Unknown	26 (21.5)	36 (19.3)		18 (18.8)	44 (20.8)		14 (22.6)	48 (19.5)	
Stroke unit length of stay, median (IQR), days	2 (2)	2 (2)	0.622	2 (3)	2 (2)	<0.976	2 (3)	2 (2)	0.549
Blood pressure									
Systolic, median (IQR), mmHg	152 (46)	160 (42)	0.052	152.5 (48)	160 (42)	0.140	151.5 (40)	160 (40)	0.251
Diastolic, median (IQR), mmHg	85 (20)	93 (25)	<0.001	85 (19)	92 (24)	0.002	87.5 (25)	90 (20)	0.040
Pulse pressure, median (IQR), mmHg	66 (34.2)	68,5 (31.2)	0.889	66 (36)	68 (31)	0.995	65.5 (36.2)	68.5 (32.2)	0.993
Biochemical markers									
NT-proBNP, median (IQR) pg/mL	864 (2556)	142 (623)	<0.001	1144.5 (2781.5)	142 (663)	<0.001	847 (2453)	199 (906)	0.001
Creatinine, median (IQR), mg/dL	0.82 (0.45)	0.82 (0.24)	0.754	0.82 (0.5)	0.82 (0.24)	0.941	0.78 (0.3)	0.83 (0.32)	0.178
Glicemia, median (IQR), mg/dL	129(87.5)	115 (49.8)	0.003	131(90.3)	115 (50)	0.004	133.5(88.8)	118 (53)	0.034

IQR, interquartile range; mRS, modified Rankin scale score; NIHSS, National Institutes of Health Stroke Scale; TOAST, Trial of Org 10,172 in Acute Stroke Treatment classification; NT-proBNP, N-terminal fragment of the prohormone of brain natriuretic peptide.

**Table 2 jpm-13-00474-t002:** Multivariate analysis of factors associated with in-hospital complications.

Variable	Multivariate Analysis Dependent Variable: All In-Hospital Complications *		Multivariate Analysis Dependent Variable: Systemic Complications ^†^		Multivariate Analysis Dependent Variable: Neurological Complications ^‡^	
	Adjusted OR ** (95% CI ***)	*p* Values	Adjusted OR (95% CI)	*p* Values	Adjusted OR (95% CI)	*p* Values
Age, years	-	-	-	-	1.027 (1.005–1.048)	0.015
NT-proBNP > 326, pg/mL	2.282 (1.269–4.104)	<0.005	2.336 (1.259–4.335)	0.007	-	-
NIHSS	1.203 (1.128–1.283)	<0.001	1.193 (1.120–1.271)	<0.001	1.090 (1.042–1.141)	<0.001
Glycaemia, g/dL	1.005 (1.001–1.009)	0.007	1.005 (1.002–1.009)	0.005	1.004 (1.000–1.007)	0.066
Diastolic blood pressure, mmHg	0.973 (0.956–0.991)	0.003	0.973 (0.955–0.992)	0.024	-	-
Mechanical thrombectomy	0.297 (0.113–0.781)	0.014	0.425 (0.168–1.073)	0.070	-	-

* Logistic regression analysis adjusted for the variables listed and the following interaction variables: sex, age, previous mRS < 2, diabetes mellitus, smoking, alcohol abuse, atrial fibrillation, coronary heart disease, renal dysfunction, prior treatment with anticoagulants, antihypertensives, NIHSS on admission, mechanical thrombectomy, stroke etiology by TOAST classification, systolic blood pressure and diastolic blood pressure. † Logistic regression analysis adjusted for the variables listed and the following interaction variables: sex, ages, previous mRS < 2, diabetes mellitus, smoking, atrial fibrillation, coronary heart disease, renal dysfunction, prior treatment with anticoagulants, treatment with insulin, mechanical thrombectomy, stroke etiology by TOAST classification and diastolic blood pressure. ‡ Logistic regression analysis adjusted for the variables listed and the following interaction variables: age, previous mRS < 2, smoking, alcohol abuse, atrial fibrillation, stroke etiology by TOAST classification and diastolic blood pressure. NT-proBNP, N-terminal fragment of the prohormone of brain natriuretic peptide; NIHSS, National Institutes of Health Stroke Scale. ** OR: Odds ratio. *** CI: Confidence interval

## Data Availability

The data presented in this study are available upon request from the corresponding author. Data are not available to the public due to personal data protection.

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
