# Peer review of "N-Terminal Pro-Brain Natriuretic Peptide Levels Are Associated with Post-Stroke In-Hospital Complications"

_jpm, 2023, doi:10.3390/jpm13030474_

Round 1

Reviewer 1 Report

Page 2. Considering the complexity of the definition of the outcomes (systemic and neurological complications), they should be clearly highlighted in a different paragraph with subheading.

Page 2, line 90. From the text it seems that the adjudication of the outcome is made by a physician that is not involved in the study, as "blinded to the conduct of this study". Please specify what does it means. If this is the case, how the Authors can be sure that the adjudication have been made consistently with the same criteria for all the patients?

Given the several outcome criteria, the adjudication should have been done by a group of dedicated physician, applying the same protocol.

Page 3. Results paragraph. Exclusion criteria should be presented in the method section.

Table 1. Please remove the template text and add the title/description of the table.

Please show the pulse pressure values, and test the significance in the regression models.

Figure 2. How the comparison between the subgroups has been performed? did the Authors used some multi comparison post hoc test? Please specify, and add the information also in the statistics paragraph.

Table 2.

- It is not clear whether the variables like "age", with no values showed in the first two models were still forced into the model but not significant, or if at the end they where excluded from the model because non significant. In the latter case, it is highly recommended that key variables like "age" or "blood pressure", or "creatinine", "AF", "ischemic heart disease" which are strongly associated (from a pathophysiological point of view) with the outcome and/or with the NT-proBNP levels, should be forced (kept) into the model even if not significant.

- How the Authors managed the variable "stroke etiology"? Indeed, it is a discrete, multiple levels, non-progressive variable, so is it quite complicated to be included in a regression model. This variable should better not be used for the regression model.

Page 7, line 224. To claim that the association of NTproBNP with the outcome is independent of age, which is crucial, the regression model should include at the same time both age and NTproBNP. This is not clear. The same is true for other confounders, see previous comment on table 2.

Reviewer 2 Report

This is a very interesting paper about N-terminal pro-brain natriuretic peptide levels are associated 2 with post-stroke in-hospital complications. Authors conducted a retrospective analysis on 308 acute ischemic stroke.

Results are very interesting and suggest NT-proBNP could be a predictor of in-hospital systemic complications.

I reccomend to cite possible genetic association with ischemic stroke (RANK/RANKL/OPG pathway: genetic association with history of ischemic stroke, Biscetti et al. 2016) as for Italian population.
